# A Methodology to Predict and Optimize Ease of Assembly for Injected Parts in a Family-Mold System

**DOI:** 10.3390/polym13183065

**Published:** 2021-09-10

**Authors:** Chao-Tsai Huang, Tsai-Wen Lin, Wen-Ren Jong, Shia-Chung Chen

**Affiliations:** 1Department of Chemical and Materials Engineering, Tamkang University, New Taipei City 25137, Taiwan; chdy1245@gmail.com; 2Department of Mechanical Engineering, Chung Yung Christian University, Taoyuan City 32023, Taiwan; wenren@cycu.edu.tw (W.-R.J.); shiachun@cycu.edu.tw (S.-C.C.)

**Keywords:** injection molding, degree of assembly, a family mold system, CAE-DOE optimization

## Abstract

In this study, the assembly behavior for two injected components made by a family mold system were investigated. Specifically, a feasible method was proposed to evaluate the characteristic length of two components within a family mold system using numerical simulation and experimental validation. Results show that as the packing pressure increases, the product index (characteristic length) becomes worse. This tendency was consistent for both the simulation prediction and experimental observation. However, for the same operation condition setting through a basic test, there were some differences in the product index between the simulation prediction and experimental observation. Specifically, the product index difference of the experimental observation was 1.65 times over that of the simulation prediction. To realize that difference between simulation and experiment, a driving force index (DFI) based on the injection pressure history curve was proposed. Through the DFI investigation, the internal driving force of the experimental system was shown to be 1.59 times over that of the simulation. The DFI was further used as the basis for machine calibration. Furthermore, after finishing machine calibration, the integrated CAE and DOE (called CAE-DOE) strategy can optimize the ease of assembly up to 20%. The result was validated by experimental observation.

## 1. Introduction

A family mold structure is one kind of the multi-cavity systems in the injection molding process. It has been utilized in the injection molding industry to make a series of assembly components for years. The related products are commonly in hand-held bar codes [1], automotive components [2], smartphone lenses [3], luggage [4], watercraft [5], toys [6], and so on. Due to various influencing factors and the complicated features of the components, the assembly behavior is quite sensitive to the design and to the processing during the injection molding. However, there is very little information to describe the relationship between the assembly behavior of injected components and the injection molding factors. Hence, it is very difficult to predict the assembly behavior in the design phase for those assembly components.

Moreover, the degree of assembly could be associated with the design for manufacturing and assembly (DFMA) standards. The main target is to integrate multiple components with multiple functions to minimize some indicators such as energy consumption, carbon footprint, number of parts, required amount of material, assembly time, and manufacturing costs [7,8,9,10]. The degree of assembly is the total of the ease of assembly indicators calculated by a series of manual handling and insertion analyses on the existing design based on the Boothroyd and Dewhurst (BD), Lucas Hall (LH), and Hitachi Assembly Evaluation (AEM) methods, among others. However, to the best of our knowledge, there is very little information to discuss the assembly behavior of injected components associated with the injection molding process. Meanwhile, some studies have discussed the degree of assembly of injected parts [11,12,13]. Unfortunately, most of them are provided as patents which are disclosed by know-how without detailed mechanism information. Hence, a determination of how to predict the sensitivity of the ease of assembly for assembling injected components from the injection molding process in the design phase has not yet been fully constructed.

Furthermore, considering the testing through the manual handling and insertion analysis on the existing design, the assembly behavior will be influenced by the final geometrical structures of the injected parts. Theoretically, if each injected component can maintain the final shape as close to the design as possible, the assembly behavior will be smooth. Therefore, the strategies developed in the literature to reduce the shrinkage and warpage of the individual injected parts might be the good solutions for the assembly behavior of family mold products. Lee and Kim [14] used the thickness of injected parts as the control factor to minimize the warpage of the product. Leo and Cuvelliez [15] focused on the gate geometry and operation parameters to modify the dimensional accuracy of the parts. Later, Yen et al. [16] selected the diameter and the length of the runner to minimize the warpage of the injected parts. Zhai et al. [17] tried to catch the balanced flow through runner size modification and then improve the product quality. They found that the target could be obtained by adjusting the runner sizes using a non-dominated sorting genetic algorithm. Othman et al. [18] conducted a study of the influence of the runner length and gate location on warpage and shrinkage to control part quality. Moreover, many researchers have moved to material modification to enhance product quality. Thomason et al. [19,20] investigated the fiber reinforced effects on the quality improvement of products experimentally. Kovacs and Solymossy [21] applied glass bead-filled PA6 to reduce the warpage and shrinkage of the injected parts. Hakimian and Sulong [22] studied different thermoplastic composites to determine the improvement on warpage. Furthermore, to enhance the quality of the injection molding parts effectively, scientists and researchers have integrated computer-aided engineering (CAE) techniques and various optimization strategies. Ozcelik and Erzurumlu [23] integrated finite element analysis, DOE, response surface methodology (RSM), and genetic algorithms to reduce warpage effectively. They claimed that after being optimized, the warpage of injected part was reduced by 51%. Zhai and Xie [24] combined CAE and sequential linear programming (SLP) to optimize the gate design to obtain a balanced flow, and then to reduce the warpage. Chiang and Chang [25] introduced RSM to optimize the shrinkage and warpage of a cell phone. They concluded the shrinkage and warpage of injected parts can be reduced by 53.9%. Fernandes et al. [26] integrated multi-objective genetic algorithms and CAE techniques to optimize the cooling channel to minimize the warpage of the injected parts. Tsai and Tang [27] utilized RSM to search for the optimal conditions to optimize the accuracy of spherical lenses. Xu and Yang [28] combined the Taguchi method, neural networks, and grey correlation analysis (GCA) to solve the multi-objective optimization problem. Kitayama et al. [29] applied a sequential approximate optimization (SAO) based on a CAE simulation to determine the optimal process parameters. They concluded that a multi-objective design optimization is effective for weld line reduction and clamping force minimization. Later, Hentati et al. [30] and Huang et al. [31] integrated CAE and the Taguchi method to optimize injection molding process parameters. In addition, Fernandes et al. [32] reviewed the studies done in the field of theoretical modeling and various optimization techniques for the injection-molding process. The strengths and weaknesses of each technique were discussed. It is noted that, in recent years, using CAE technology to perform injection molding simulations virtually can enhance efficiency in the product development and problem solving effectively. However, it is quite common to encounter some difference between simulation predictions and experimental observation. Huang et al. [31,33] proposed a feasible method to discover the cause of that difference. They concluded that to diminish that difference, both the virtual and real injection molding machines should be calibrated. Moreover, to discuss the performance of injection molding machines, Chen et al. [34,35] proposed a method to derive the correlation between product quality and machine quality indexes. They concluded that the pressure peak index, viscosity index, and energy index are strongly associated with product quality. Some quality index could be useful to discover the internal driving force in the future.

As described above, it is noted that due to the non-balanced component structure, the retention of dimensional precision for the individual components made by a family mold system is strongly affected by operation conditions. Moreover, which interaction between the individual component forming which will further influence assembly behavior is not clear. Hence, in this study, the assembly behavior of two injected components made by a family mold system was investigated. Specifically, a method to evaluate the characteristic length of two components within a family mold system is proposed using numerical simulation and experimental validation. Then, the correlation between the characteristic length and the ease of assembly is further discussed. Moreover, to enhance the simulation’s accuracy of the assembly behavior, machine calibration was performed. The influence of machine calibration on the assembly behavior is then discussed. Furthermore, to optimize the ease of assembly for this complicated system, the integrated CAE and DOE (called CAE-DOE) is utilized for virtual optimization. Then the optimization efficiency in the assembly behavior is verified by the physical DOE experiment.

## 2. Theory and Assumption

The polymer material in this study can be assumed to be a general Newtonian fluid (GNF). During the injection molding process, the non-isothermal 3D flow motion can be mathematically described by the following equations:(1)$$\frac{\partial \rho }{\partial t}+\nabla \cdot \rho u=0\;$$
(2)$$\frac{\partial }{\partial t}\left(\rho u\right)+\nabla \cdot \left(\rho uu-\sigma \right)=\rho g\;$$
(3)$$\sigma =-pI+\eta \left(\nabla u+\nabla u^{T}\right)\;$$
(4)$$\rho C_{p}\left(\frac{\partial T}{\partial t}+u\cdot \nabla T\right)=\nabla \cdot \left(k\nabla T\right)+\eta \dot{\gamma }\;$$
where **u** is the velocity vector, *T* is the temperature, *t* is the time, *p* is the pressure, **σ** is the total stress tensor, *ρ* is the density, **g** is gravitational force, **I** is the unit matrix, *η* is the viscosity, *k* is the thermal conductivity, *C_p_* is the specific heat, and $\dot{\gamma }$ is the generalized shear rate.

Moreover, the modified-cross model with Arrhenius temperature dependence is employed to describe the viscosity of polymer melt:(5)$$\eta \left(T,\dot{\gamma }\right)=\frac{\eta_{o}\left(T\right)}{1+{\left(\eta_{o}\dot{\gamma }/\tau^{*}\right)}^{1-n}}$$
where
(6)$$\eta_{o}\left(T\right)=BExp\left(\frac{T_{b}}{T}\right)$$
where η is the viscosity, $\eta_{o}$ is the zero shear viscosity, *n* is the power law index, *B* is the consistency index, and $\tau^{*}$ is the parameter that describes the transition region between the zero shear rate and the power law region of the viscosity curve.

## 3. Methodology and Materials

In this study, a numerical simulation and experimental methods were utilized. The associated systems for both methods are described as follows.

### 3.1. Numerical Simulation System

Regarding the numerical simulation of the system, Moldex3D R16^®^ software (supplied by CoreTech System Co. Ltd., Hsinchu County, Taiwan) was adopted. The geometrical structure of the system is shown in Figure 1. Specifically, it is a family mold system with two components of parts A and B, as shown in Figure 1a. Part A is an inner part, and part B is an outer part. The associated runner, with detailed dimensions, is listed in Figure 1b. In addition, the dimensions of part A and part B are exhibited in Figure 1c,d. Their dimensions are about 40 mm × 40 mm × 14 mm. In addition, the volumes of part A and part B were 5.9 and 6.5 cm^3^, respectively. Furthermore, the moldbase and cooling channel layout are presented in Figure 2. There are four cooling channels inside the moldbase. For injection molding, the material used was acrylonitrile butadiene styrene, called ABS (PA757 supplied by Che-Mei, Tainan city, Taiwan). In order to perform the injection molding simulation, several material properties need to be measured and stored as the database. For example, the key properties to influence the flow and warpage are the temperature-shear rate-dependent viscosity and the specific volume against pressure-temperature (pvT), as presented in Figure 3. Those data were measured and provided from Moldex3D directly. Furthermore, to evaluate the assembly behavior of these components, and to find out the key practical factors for further study, a single factor test was performed with the associated factors as listed in Table 1. Specifically, each factor has five levels. The reasons we selected those factors is referred to in several studies mentioned earlier [14,15,16,17,18,21,22,23,24,25,27,28,29,30,31]. The key operation parameters utilized in each reference are listed in Table 2. The goal of the single factor test was to find out some practical operation parameter which can be utilized as the major control factor to evaluate the variation of the characteristic lengths. Those lengths were used to evaluate the assembly behavior and will be explained later.

Moreover, a basic test of the injection molding simulation was performed to determine the relationship between the characteristic lengths and the assembly behavior. The operation conditions for the basic test are listed in Table 3. Specifically, the melt temperature was 210 °C. The mold temperature was 50 °C. The injection speed was setup at 50% which was based on the maximum speed of the screw movement, with 125 mm/s in the machine (afterwards called injection speed 50% setting). The packing time was 7 s. The cooling time was 11 s. The packing pressure was setup from 25% to 100%. Here the packing pressure setting was based on the end of filling pressure, P_EOF_.

### 3.2. Experimental Equipment

Moreover, to discover the real variation of the assembly behavior based on the characteristic lengths and to validate the simulation predictions, an injection molding experiment was constructed as follows. Figure 4a presents the FCS 150SV injection machine supplied by Fu Chun Shin Machinery Co. Ltd., Tainan City, Taiwan. This system offers a maximum injection pressure of 140 bar, a maximum injection speed of 125 mm/s (the maximum speed of the screw movement, as mentioned previously), and a maximum movement distance of 200 mm for the screw. The screw diameter is 44 mm. In addition, the real mold structure is listed in Figure 4b. The dimensions for the cavity, runner, and cooling channels are as described in Figure 2.

### 3.3. Define the Characteristic Length as Product Index

To study the assembly behavior, some characteristic lengths are presented as in Figure 5. Specifically, the product index based on the characteristic length is defined as follows:X_i_ = (X_Bi_ − X_Ai_)(7)
where i is from 1 to 4; X_Ai_ is the outer length of part A, and X_Bi_ is the inner length of part B.

For example, X_1_ = (X_B1_ − X_A1_) is the characteristic length at a central location (based on part A), obtained from the difference between the inner length of part B and the outer length of part A on the top plane. X_2_ = (X_B2_ − X_A2_) is the characteristic length at the end location of the top plane (based on part A). Similarly, X_3_ and X_4_ are defined on the bottom plane. Theoretically, when the characteristic length (Xi) is greater than zero where the inner length of part B is larger than the outer length of part A, the assembly should be easy. On the other hand, if the characteristic length (Xi) is smaller than zero, the assembly should be not easy.

### 3.4. Integrate CAE and Design of Experiments (DOE) to Optimize the Key Factors in the Assembly Behavior

The operational parameters of injection molding that influence the assembly behavior of injected parts are very complicated. To examine the influence of the key factors on the assembly behavior and to optimize them, design of experiment (DOE) optimization was introduced. Specifically, DOE methods based on CAE technology (afterwards called CAE-DOE) and physical DOE experiments were utilized. Here, to discover the optimization efficiency of the DOE method before performing the machine calibration, the key control factors include (A) injection speed, (B) mold temperature, (C) packing pressure, (D) packing time, (E) melt temperature, and (F) cooling time, as listed in Table 4. For each factor, three levels have been specified. For example, regarding the injection speed factor, level 1 to level 3 is 25 mm/s (20% injection speed setting) to 125 mm/s (100% injection speed setting). Before discussing efficiency, all the operation parameters of the level 2 column were selected as the original design setting for CAE-DOE investigation. The characteristic lengths of the injected parts based on this original design setting were used as the basis for further comparison. In addition, the corresponding orthogonal array for DOE performance using CAE (i.e., CAE-DOE) is listed in Table 5. Indeed, eighteen sets of injection molding trials was executed numerically based on the L_18_ (2^1^ × 3^7^) orthogonal array. Since only six major factors were considered and each factor has three levels, the first and the eighth columns will be ignored for further application.

The eighteen sets of injection molding trials were executed. The associated characteristic lengths and their average for each set were measured and will be discussed later, in the Results and Discussion section. Moreover, based on the calculated characteristic lengths for each set, the standard deviation Sn was calculated from Equation (8). Then the S/N ratio (signal-to-noise ratio) was obtained by Equation (9).
(8)$$S_{n}\;(\mathrm{Standard}\;\mathrm{Deviation})=\sqrt{\frac{\sum {\left(y_{i}-\bar{y}\right)}^{2}}{N-1}}$$
(9)$$S/N=-10\;\times \;\log[{\left(\bar{y}-y_{0}\right)}^{2}+{S_{n}}^{2}]$$
where for each injected part: $y_{i}$ is the deviation between the *i*th characteristic length; $\bar{y}$ is the average of four characteristic length deviations; and *y*_0_ is 0.

## 4. Results and Discussion

### 4.1. Single Factor Test and Basic Test for Assembly Behavior

#### 4.1.1. Perform a Single Factor Test

As mentioned earlier, the purpose of the single test was to determine some practical operation parameters for further study of the assembly behavior. Figure 6 shows the results of the single factor test on the influence of the characteristic lengths. For each factor, there were five levels to test, as listed in Table 1. The deviation was estimated by the maximum value of the characteristic length minus the minimum value, at each location. Then, the average deviation was utilized to evaluate the sensitivity of each factor. For example, in Figure 6a, when the injection speed setting increased from 30% to 70%, the average deviation from the injection speed was about 0.023 mm. For the melt temperature effect, the average deviation was 0.065 mm, as seen in Figure 6b. Figure 6c presents data that the mold temperature does not provide significant influence, with only a 0.001 mm deviation. Moreover, when the packing pressure is increased, it provides a significant influence, with an average deviation of 0.121 mm, as shown in Figure 6d. Similarly, the average deviations for the packing time and cooling time effects were 0.046 mm and 0.058 mm, respectively, as shown in Figure 6e,f. Overall, the packing pressure effect had the most significant influence on the variation of the characteristic lengths. In addition, the variation tendency of the characteristic lengths was almost proportional to the changes of the packing pressures. Hence, the packing pressure effect was selected as the practical parameter for further study of the assembly behavior.

#### 4.1.2. Perform a Basic Test

The goal for the basic test was to understand the flow behavior and the shrinkage behavior of each location for parts A and B. It was also used to realize the correlation between the characteristic length and the assembly behavior through simulation prediction and experimental verification when the injection molding simulation was performed using the operation condition of Table 3 at a 50% packing pressure setting. In Figure 7, when the volume was filled at 37.5%, the flow behavior for both parts A and B looked similar. However, from 63% to 100% volume filled, the flow imbalance phenomenon happened as expected due to the volume difference of cavities A and B. Figure 8 shows the shrinkage behavior for parts A and B. Regarding part A, X_A1_ and X_A3_ shrunk significantly because there was no constraint. The higher packing pressure, the worse the shrinkage happened, as shown in Figure 8c. However, since X_A2_ and X_A4_ were located at the end portion with strong wall constraints, the higher packing pressure provided the expansive result. On the other hand, for part B, since X_B1_ and X_B2_ were located within a concrete plane, when the packing pressure was increased, their lengths increased slightly. At the same time, X_B3_ and X_B4_ shrunk significantly because of lack of any constraint.

Furthermore, since the variations of the individual lengths for parts A and B were not in the same trend, we had to define the characteristic length to measure the interaction among those individual lengths. In Figure 9a, there were four tests with different packing pressure settings for simulation prediction. As the packing pressure increased, the variation of the characteristic lengths was almost linearly changed. To give better understanding, two packing pressures with lower and higher settings were selected and presented in Figure 9b. For example, at a 25% (lower) packing pressure setting, the characteristic lengths on the top plane were less than zero (X_1_ = −0.207 mm and X_2_ = −0.224 mm). Also, the characteristic lengths on the bottom plane were close to or less than zero (X_3_ = 0.004 mm and X_4_ = −0.013 mm). This means that the inner lengths of part B were less than outer lengths of part A. Theoretically, these components are not easy to assemble. When the packing pressure was increased to the 100% (higher) packing pressure setting, the characteristic lengths on the top plane were far less than zero (X_1_ = −0.175 mm and X_2_ = −0.255 mm). Also, the characteristic lengths on the bottom plane were less than zero (X_3_ = −0.061 mm and X_4_ = −0.083 mm). When the higher packing pressure was applied, although X_1_ became more positive, the others (X_2_ to X_4_) became more negative. This led the inner lengths of part B to become much smaller than outer lengths of part A, theoretically resulting in a more difficult assembly of parts A and B. Indeed, higher packing pressure is not a solution to manage the degree of assembly in this study.

A basic evaluation of the assembly behavior for parts A and B is also performed experimentally. The operation conditions were the same as mentioned in the basic numerical test (see Table 3). For each packing pressure operation, six samples each of part A and part B were collected to measure the inner lengths and outer lengths, as described in Figure 5 and Equation (7). Then, the associated average characteristic lengths were obtained and plotted, as in Figure 9c. As the higher packing pressure was applied, the X_1_ became more positive, and the others (X_2_ to X_4_) became more negative. The basic tendency is similar to that of the numerical prediction. To get a better understanding, two packing pressures with lower and higher settings were selected and presented in Figure 9d. At the 25% (lower) packing pressure setting, the characteristic lengths on the top plane were less than zero (X_1_ = −0.035 mm and X_2_ = −0.24 mm). Also, the characteristic lengths on the bottom plane were also less than zero (X_3_ = −0.007 mm and X_4_ = −0.145 mm). When the packing pressure was increased to the 100% (higher) packing pressure setting, the characteristic lengths on the top plane changed. X_1_ becomes more positive but X_2_ became far less than zero (X_1_ = 0.042 mm and X_2_ = −0.267 mm). The characteristic lengths on the bottom plane were also far from zero (X_3_ = −0.167 mm and X_4_ = −0.280 mm). This means that the higher packing pressure will lead the inner lengths of part B to become much smaller than the outer lengths of part A, resulting in more difficulty in the assembly of parts A and B. Clearly, the tendency of the change of the characteristic lengths is in reasonable agreement for both the simulation prediction and the experimental measurement.

#### 4.1.3. Correlation between Characteristic Length and Assembly Behavior

To realize the relationship between the characteristic lengths and the assembly behavior, a real integration test for parts A and B was performed, as shown in Figure 10. At the 25% packing pressure setting, the integration process assemble parts A and B was smooth and without difficulty. From the top view and side view, it is clearly seen that the assembly of parts A and B was completed as shown in Figure 10a. However, when the packing pressure setting was changed to 100%, the integration test for the assembly became very difficult. The integration test failed. Practically, the higher the packing pressure utilized, the more difficulty encountered in the assembly operation. The results of the integration test are consistent with the characteristic length behavior of the simulation prediction and experimental observation, as discussed previously. Obviously, this method to evaluate the assembly behavior using the characteristic lengths is feasible qualitatively so far.

### 4.2. Discover the Reason for the Difference between Simulation and Experiment for the Assembly Behavior

Figure 11 shows the comparison between the characteristic lengths of the simulation prediction and those of experimental measurement. When the packing pressure increased from 25% to 100%, X_1_ increased, while the others (X_2_ to X_4_) decreased for both the simulation and the experiment. However, when the comparison proceeded one-by-one from X_1_ via X_2_ to X_4_, the amounts of simulation prediction were under-predicted at the top plane for X_1_ and X_2_ (i.e., too much negative), and were over-predicted at bottom plane for X_3_ and X_4_ (i.e., too much positive). Overall, the tendency is in reasonable agreement, but the amount of characteristic length at each location was not exactly matched in both the simulation and the experiment. To discover the difference between the simulation and the experiment, the relationship between the internal driving force from the injection machine and the characteristic length difference (∆Xi) on the injected parts was investigated. Here, the characteristic length difference (∆Xi) is defined as Equation (10), as follows.
∆X_i_ = Xi (at 100% Packing) − Xi (at 25% Packing)(10)
where i is from 1 to 4.

For example, ∆X_1_ (the characteristic length difference of the simulation) is equal to 0.032 mm (i.e., (−0.175) − (−0.207) = 0.032 mm). Other characteristic length differences of the simulation were calculated, and are listed in Table 6. The average characteristic length differences of the simulation were further calculated and are listed in the rightmost column in Table 6. The ∆X_1_ of the experiment was equal to 0.077 mm (i.e., (0.042) − (−0.035) = 0.077 mm). The other characteristic length differences and their average were also calculated and are shown in Table 6. It is noted that the average of the characteristic length differences of the experiment was about 1.65 times over that of the simulation prediction (that is, 0.061/0.037 = 1.65). For the exact same operation condition settings for both simulation and experimental systems, why did the experimental system drive more dimensional variation in the final injection parts than its simulation counterpart is a very interesting question. Before we proceed to answer this question, we note that Huang et al. [33] mentioned that one of the reasons for the difference between numerical predictions and experimental observations in injection molding is because the real machine (experiment) and the virtual machine are not the same, even they have the same operation condition settings. To reduce the difference between two systems, some injection machines need to be calibrated. The details will be discussed in the following section.

### 4.3. Machine Calibration Effects on Assembly Behavior

#### 4.3.1. Perform Machine Calibration

To reduce the difference between the numerical prediction and the experimental observations, machine calibration procedures were performed based on the direction in [33]. However, the new controller was changed into the FCS injection machine and one pressure transducer was installed at gate location. Some calibration procedures needed to be modified from the the system in [33]. Specifically, the reference point to catch the injection pressure history curve was moved to the gate location; the new controller was installed, and the injection pressure history of the experiment was different. The details of the calibration procedures are as follows. Machine calibration is based on the injection pressure history curves for both the simulation and the experiment, using a circle plate system with a pressure transducer installed at the gate. The sensor location is presented in Figure 12. The data of the injection pressure can be recorded from the pressure transducer at the gate, which is further used to create the pressure history curve for machine calibration. Specifically, for the same operation condition settings, Figure 13a shows the injection pressure history curves for both the simulation and the experiment at the 50% injection speed setting. The experiment shows a higher injection pressure history curve than that of the simulation counterpart over the entire period, which means that the real injection machine has a higher driving force than that of the virtual simulation machine. To evaluate the driving force for the injection molding through the entire cycle, a driving force index (DFI) based on the total accumulated driving force was defined, as in Equation (11). It is also called the viscosity index [34,35]. This equation can be regarded as reflecting the accumulated resistance force of melt flow during injection molding.
(11)$$\mathrm{DFI}={\langle P_{Total}\rangle }_{i}=\int_{0}^{t}{\left(P_{inj}\right)}_{i}\;dt$$
where *i* is either simulation or experiment, $\langle P_{Total}\rangle$ is the total accumulated driving force with (MPa·s) viscosity units, and *P_inj_* is the injection pressure measured at the gate location at the time *t*.

Through this equation, the total accumulated driving force is equal to the integration area under the injection pressure history curve for both the simulation and experiment systems. For instance, at the 50% injection speed setting, the DFI of the real experimental system, $\langle P_{Total}\rangle {}_{Exp}$, was about 1471.6 MPa·s, and the DFI of the virtual simulation system, $\langle P_{Total}\rangle {}_{Sim}$, was about 928.3 MPa·s, as shown in Figure 13b. Specifically, the DFI of the real experimental system was about 1.59 times over that of the virtual simulation system. This result is quite consistent with that ratio of 1.65 times for the experimental product index (characteristic lengths) difference from the simulation, as described previously. Based on this idea, the key to calibrating the machine is to determine the matched pair for both the simulation and the experimental systems that have the same DFI for the injection molding. For example, when the real injection machine keeps the 50% injection speed setting, the curve of the simulation system with the 50% injection speed setting is lower (i.e., with lower driving force). At this moment, the injection speed setting can be increased to enhance the driving force virtually. Until the injection speed setting of the simulation is increased to 110%, the injection pressure history curve is very close to that of the real injection machine with the 50% injection speed setting, as shown in Figure 13d. In this Figure, $\langle P_{Total}\rangle {}_{Exp}$ is about 1471.6 MPa·s, and $\langle P_{Total}\rangle {}_{Sim}$ is about 1449.2 MPa·s. The DFI of the real experimental system was about 1.02 times over that of the virtual simulation system. That is to say, the internal driving force of the 50% injection speed setting experimentally was matched by the 110% injection speed setting numerically. Specifically, when the injection speed setting is 50% in the experimental study, the counterpart in the simulation would be the 79.5 mm/s injection speed setting. Other matched pairs were evaluated, and are listed in Table 7.

#### 4.3.2. Evaluate Calibration Effect on Assembly Behavior

After performing the machine calibration at various injection speed settings, the machine calibration effect on the characteristic length changes could be further examined. For example, Figure 14 presents the comparison of the characteristic lengths between the simulation and experiment at the 25% to 100% packing pressure settings before and after machine calibration at the 50% injection speed setting. Since those four different characteristic lengths have different variation behavior, the calibration effect on each characteristic length was calculated individually. The calibration rate for each characteristic length is defined as in Equation (12):Calibration rate = [ΔL − (ΔL)cal]/ΔL∗100%(12)
where ΔL is the difference between the experimental characteristic length and that of the simulation before machine calibration, and (ΔL)cal is the difference between the experimental characteristic length and that of the simulation one after machine calibration.

For example, when the 100% packing pressure setting situation is considered, the calibration rates for each characteristic length are shown as in Table 8. The average calibration rate is about 18%. This demonstrates that the difference between simulation prediction and experimental observation was reduced by 18%. In addition, the details of the machine calibration effect for the 50% injection speed setting at various packing pressure settings are listed in Table 9. It is noted that the calibration effect on the characteristic lengths improved about 10%. Moreover, after the machine calibration was completed, the relationship between the characteristic lengths and the assembly behavior was further measured using the integration test described in Figure 10. The result of the integration test is shown in Table 10. Specifically, when it was at the 25% packing pressure setting, although all X_i_ were smaller than zero, the real components A and B could be integrated together smoothly. Similarly, all components passed the integration test from the 25% to the 75% packing pressure settings. At the 100% packing pressure setting, the integration test failed. At this moment, it could be found that as long as one characteristic length is smaller than −0.25 mm, it would fail the integration test. Similarly, the virtual criteria to assure the good assembly of parts A and B can be obtained when one characteristic length is not smaller than −0.243 mm numerically, as listed in Table 11. Based on these results, using numerical simulation to predict the ease of assembly has been verified as a feasible and quantitative method.

### 4.4. Optimize the Assembly Behavior Using CAE-DOE

#### 4.4.1. Optimization before the Machine Calibration

Before performing the machine calibration, the eighteen sets of injection molding trials were executed using the parameters and settings found in Table 4 and Table 5. The associated characteristic lengths and their average for each set were measured are and recorded in Table 12. For example, after the first molding simulation based on Set 1 conditions, the individual characteristic lengths were −0.15, −0.25, −0.08, and −0.12 mm, respectively. Also, the average characteristic length was −0.15 mm. Based on these calculated characteristic lengths, the standard deviation Sn was 0.06 mm, which was calculated from Equation (8). Then S/N ratio (signal-to-noise ratio) was 15.82 (obtained by Equation (9)). The quality values of the remaining seventeen sets are listed in Table 12.

Moreover, before performing the machine calibration, the response for each factor was estimated and recorded into Table 13. The responses of all factors can be plotted as shown in Figure 15. From Table 13 and Figure 15, the optimized parameter set was determined as (A2, B3, C1, D2, E1, and F3). This optimized parameter set was applied in the injection molding simulation. The result is shown as “CAE-DOE (Sim)” in Figure 16. Compared to the original design, the optimized conditions reduced the average characteristic length from −0.169 mm (original) to −0.151 mm for the numerical simulation. Obviously, using the virtual DOE method (CAE-DOE), the quality can be improved about 10.7%. Moreover, to validate the efficiency of CAE-DOE optimization before performing the machine calibration, both the original design and the optimized parameter sets were utilized to execute the injection molding experimentally, and these results are also exhibited in Figure 16. The average characteristic length of the original design for the experimental system was −0.150 mm. Using the (CAE-DOE) optimized parameter set to perform the real injection molding, the average characteristic length was reduced to −0.143 mm, as demonstrated as “CAE-DOE (Exp)” in Figure 16. Clearly, the ease of assembly improved about 5%.

#### 4.4.2. Optimization after the Machine Calibration

After the machine was calibrated, the control factors and their levels were modified as listed in Table 14. The corresponding orthogonal array for DOE performance using CAE (i.e., CAE-DOE) is the same as that listed in Table 5. Then, eighteen sets of injection molding trials were performed. The associated characteristic lengths and their average for each set were measured and recorded into Table 15. The quality values of the eighteen sets are also listed in Table 15. Moreover, in the presence of the machine calibration effect, based on the S/N ratio, the response for each factor was estimated, as recorded in Table 16. The responses of all factors were plotted, as shown in Figure 17. From Table 16 and Figure 17, after performing the machine calibration, the optimized parameter set obtained was (A2, B3, C1, D2, E2, and F3). The optimized parameter set was used in the injection molding simulation. The result is demonstrated as “CAE-DOE with calibration (Sim)” in Figure 16. Compared to the original design, the optimized conditions reduced the characteristic length significantly from −0.169 mm (original) to −0.134 mm in the numerical simulation. After the machine calibration was performed, using CAE-DOE, the assembly behavior improved about 20.7% in the simulation system. Obviously, these results are consistent with those of the simulation prediction. Moreover, the efficiency of CAE-DOE optimization after machine calibration has been validated as well. Specifically, after the machine was calibrated, the average characteristic lengths of the injected parts, based on the optimized parameter set, was reduced significantly from −0.150 mm (original) to −0.119 mm in the experimental system. In addition, the real experimental validation through the integration test was performed, as shown in Figure 18. Obviously, after the machine was calibrated, the assembly behavior improved about 20.7% in the experimental system. Overall, the driving forces to improve the ease of assembly were quite consistent for both the simulation prediction and experimental observation. Moreover, both the simulation and the experimental systems benefitted from the machine calibration effect. The contribution of machine calibration to the ease of assembly is described in Figure 16. First, from the simulation point of view, before and after machine calibration the average characteristic by CAE-DOE went from −0.151 mm to −0.134 mm. The contribution of the machine calibration effect was to enhance the ease of assembly by 11.3% in the simulation prediction. Moreover, from the experimental point of view, before and after the machine calibration the average characteristic by real injection went from −0.143 mm to −0.119 mm. The calibration effect enhanced the ease of assembly by 16.8% in the real injected observation.

## 5. Conclusions

In this study, we proposed a feasible method to predict assembly behavior using the characteristic length as the product index for two components within a family mold system, using numerical simulation and experimental observation. Several key points can be obtained, as follows:(1)For the same operation condition settings of simulation and experimental systems, as the packing pressure is higher, the assembly behavior based on the characteristic lengths becomes poorer. The trend is consistent for both simulations and experiments, but there is some difference between the simulation and experimental results.(2)Based on the characteristic length variation (product index difference) investigation, under the same operation condition setting, the product index difference of the experimental observation was 1.65 times over that of the simulation prediction. Through the DFI investigation, the internal driving force of the experimental system was 1.59 times over that of the simulation one. This shows the internal driving force is quite matched with the product quality index. It also demonstrates that the simulation and experimental systems are not the same. Hence, the injection machine needs to be calibrated.(3)After the injection machine was calibrated, the criteria for good assembly based on the integration test could be constructed. Specifically, the individual characteristic lengths should be not smaller than −0.250 mm in the real system (or not smaller than −0.243 mm in the virtual simulation system). The consistency was good.(4)To handle complex injection molding processing, the CAE-DOE optimization method was verified with high efficiency in ease of assembly improvement. Moreover, after finishing the machine calibration, the improvement of the CAE-DOE optimization method could approach 20%. In addition, the driving forces to improve the assembly behavior were quite consistent for both the simulation prediction and the experimental observation. To handle the huge parameter operation window and optimize the assembly behavior, the CAE-DOE optimization strategy was applied. After finishing the machine calibration, the CAE-DOE strategy could optimize the ease of assembly up to 20%. The result is validated by experimental observation.

## Figures and Tables

**Figure 1 polymers-13-03065-f001:**
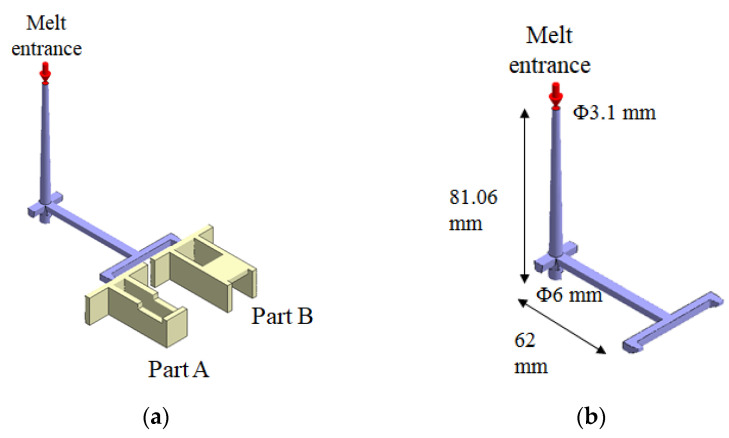
Geometrical structures of two components in a family mold: (**a**) the runner and cavities, (**b**) the dimensions of the runner, (**c**) the dimensions of part A, (**d**) the dimensions of part B.

**Figure 2 polymers-13-03065-f002:**
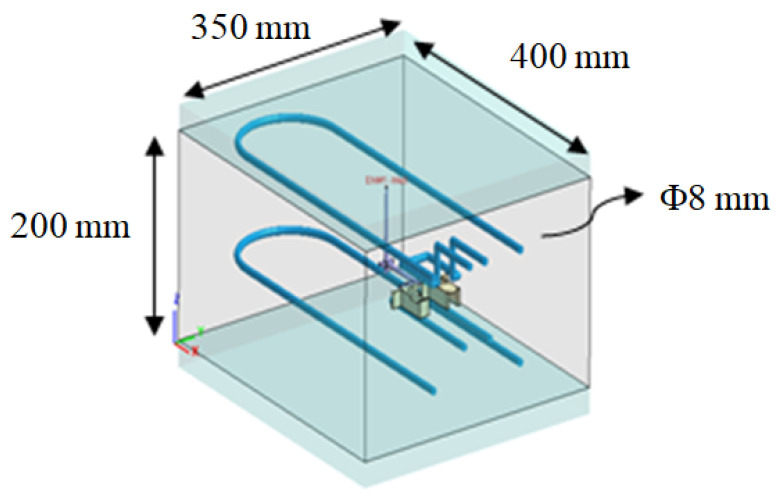
The moldbase and cooling channel layout.

**Figure 3 polymers-13-03065-f003:**
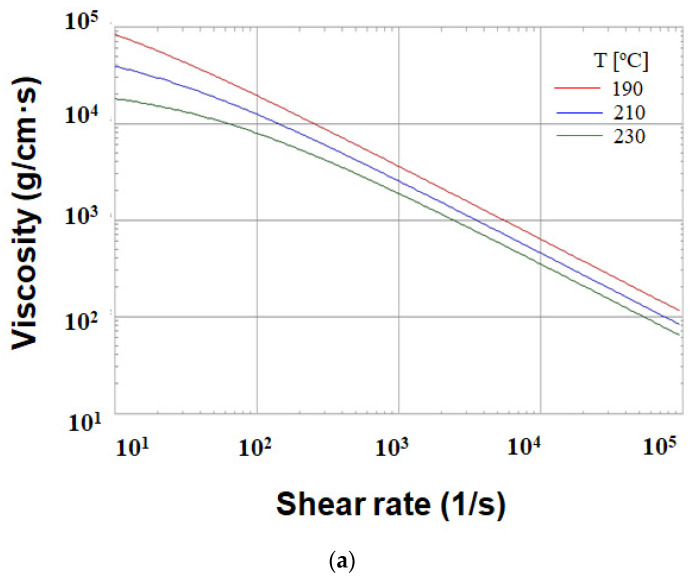
The material properties of ABS (PA757, supplied by Che-Mei): (**a**) viscosity, (**b**) pvT (the specific volume against pressure-temperature).

**Figure 4 polymers-13-03065-f004:**
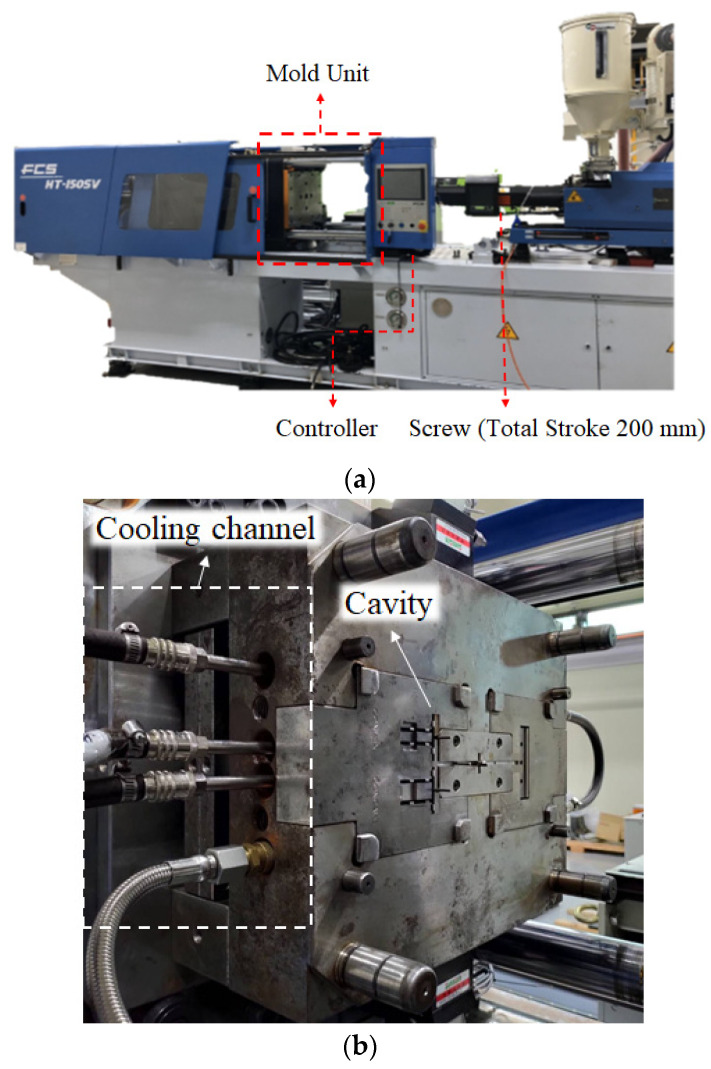
Machine and equipment setup: (**a**) injection machine (Fu Chun Shin Machinery Co. Ltd., Model: FCS 150-SV), and (**b**) mold layout.

**Figure 5 polymers-13-03065-f005:**
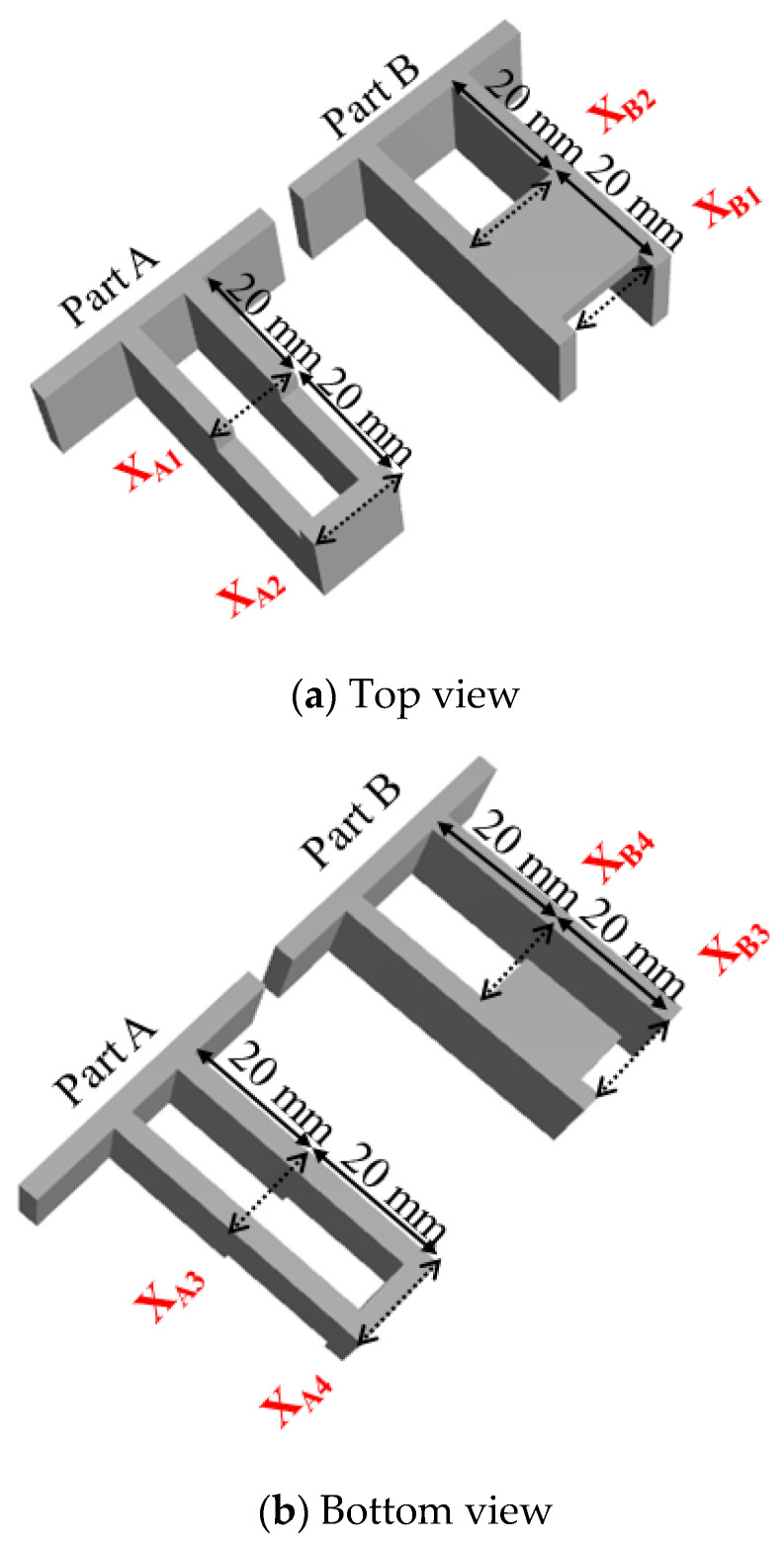
Definition of the characteristic lengths for parts A and B: (**a**) top view, (**b**) bottom view.

**Figure 6 polymers-13-03065-f006:**
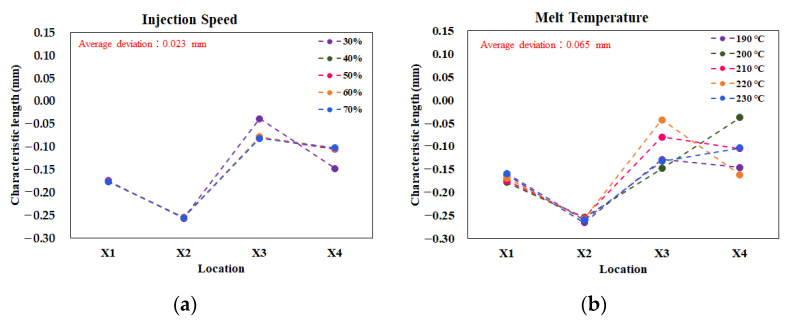
Characteristic lengths variation due to the single factor test: (**a**) injection speed, (**b**) melt temperature, (**c**) mold temperature, (**d**) packing pressure, (**e**) packing time, (**f**) cooling time.

**Figure 7 polymers-13-03065-f007:**
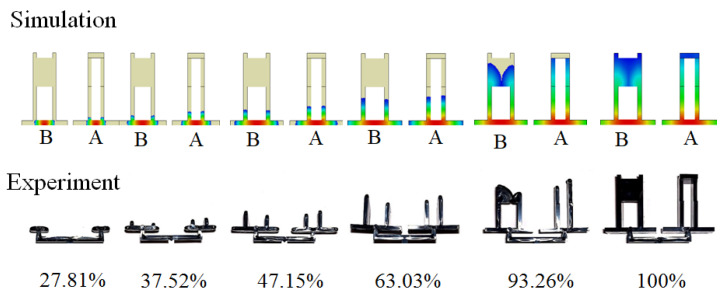
The flow behavior for both the simulation prediction and experimental validation.

**Figure 8 polymers-13-03065-f008:**
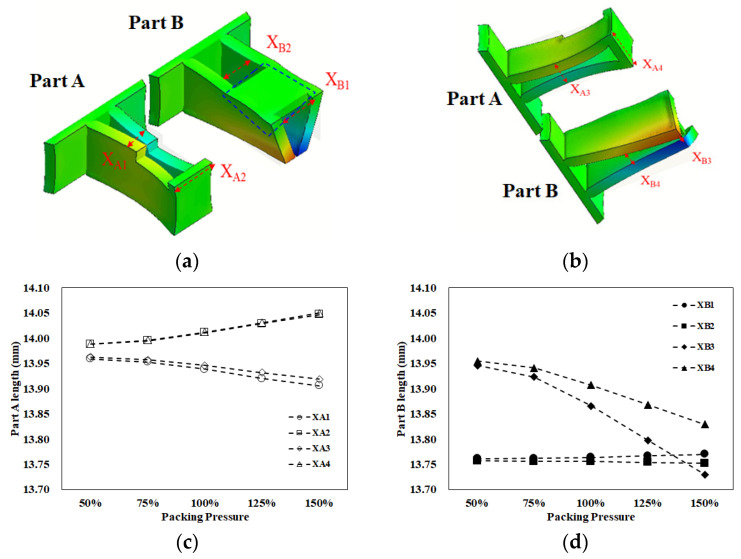
The shrinkage behavior for parts A and B: (**a**) top view, (**b**) bottom view, (**c**) individual length variation of part A, (**d**) individual length variation of part B.

**Figure 9 polymers-13-03065-f009:**
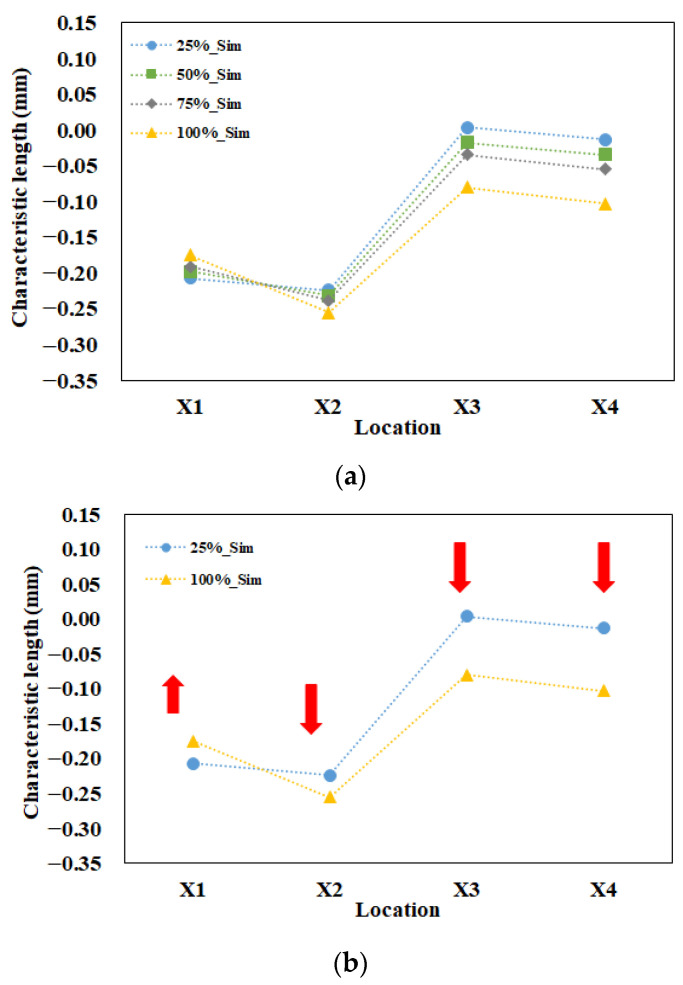
Evaluation of the ease of assembly for parts A and B: (**a**) full range of packing pressure effect for simulation prediction, (**b**) higher and lower packing pressure effect for simulation prediction, (**c**) full range of packing pressure effect for experimental measurement, (**d**) higher and lower packing pressure effect for experimental measurement.

**Figure 10 polymers-13-03065-f010:**
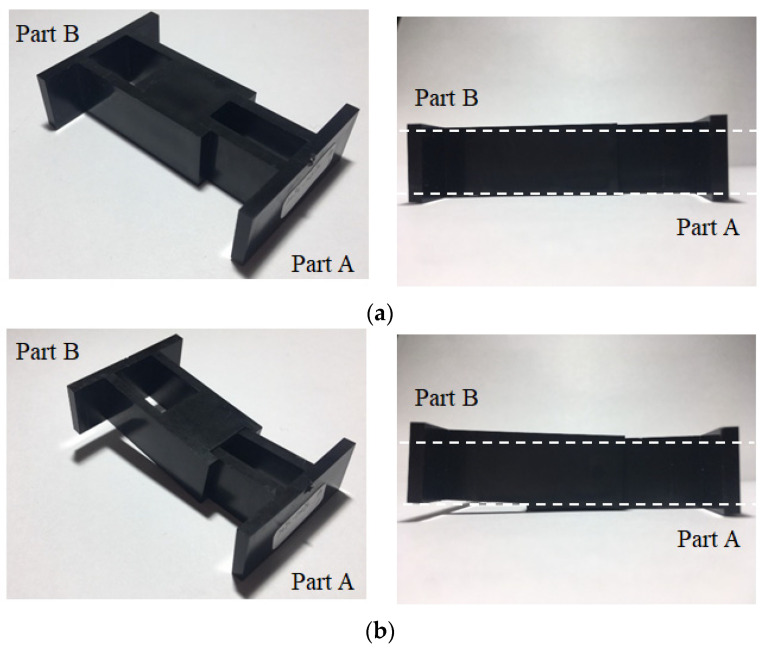
Evaluation on the degree of assembly difficulty through a real integration test for different packing pressure settings on the injection parts: (**a**) 25% packing: passed, (**b**) 100% packing: failed.

**Figure 11 polymers-13-03065-f011:**
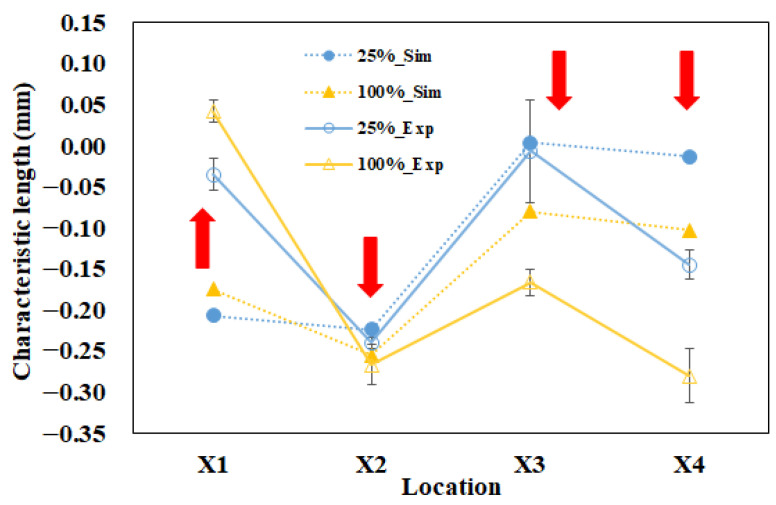
Comparison between simulation and experiment at 25% to 100% packing pressure settings.

**Figure 12 polymers-13-03065-f012:**
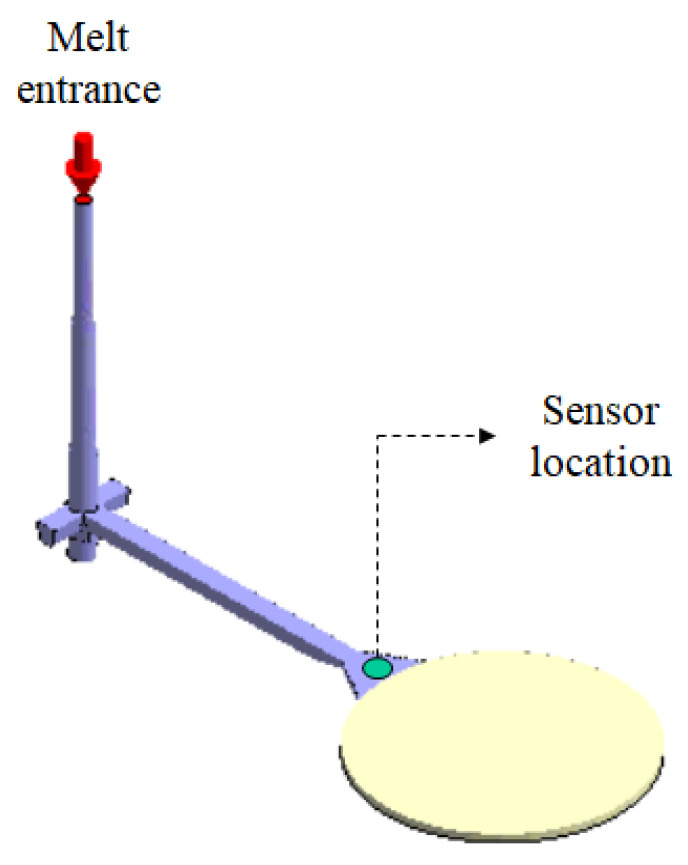
The calibration system and the sensor location.

**Figure 13 polymers-13-03065-f013:**
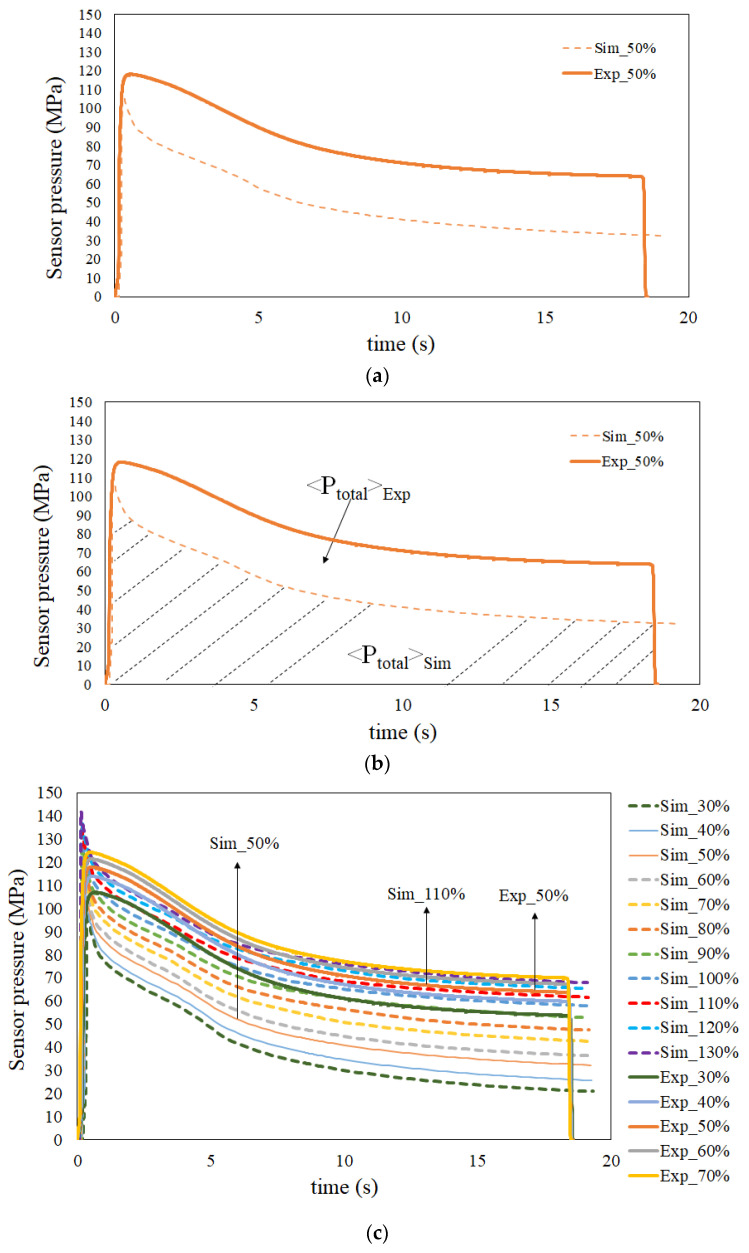
Injection pressure history curves used to perform machine calibration: (**a**) original injection pressure history curves at the 50% injection speed settings for both simulation and experiment, (**b**) schematic plots for the total driving force of the real experimental and simulation systems, (**c**) comparison of the history of injection pressure between the simulation and experiment at various injection speeds from 30% to 130%, (**d**) the matched pair for both the simulation and the experiment, where the simulation 110% injection speed setting is matched with the experimental 50% injection speed setting.

**Figure 14 polymers-13-03065-f014:**
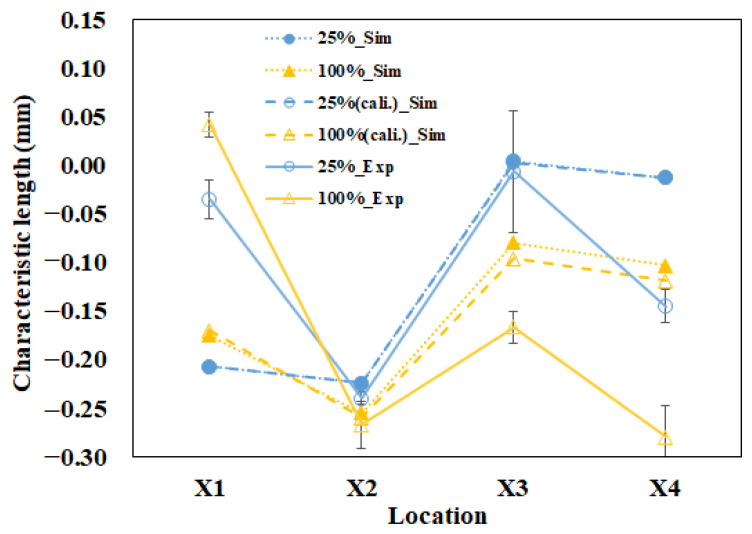
At the 50% injection speed setting, the comparison of the characteristic length deviation between simulation and experiment at the 25% to 100% packing pressure settings before and after machine calibration.

**Figure 15 polymers-13-03065-f015:**
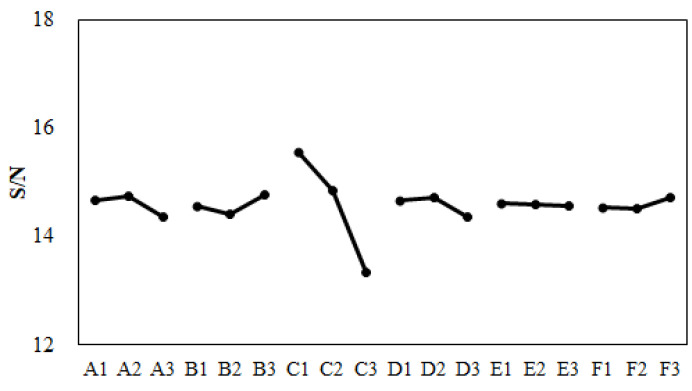
Response plot for various control factors before machine calibration.

**Figure 16 polymers-13-03065-f016:**
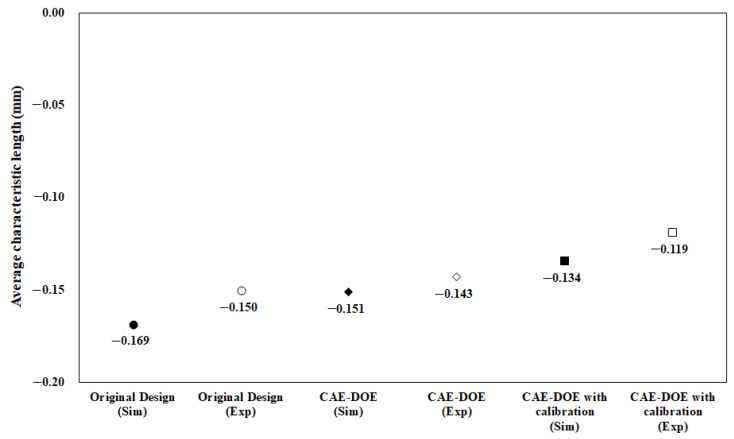
Average of the characteristic length quality change through DOE optimization.

**Figure 17 polymers-13-03065-f017:**
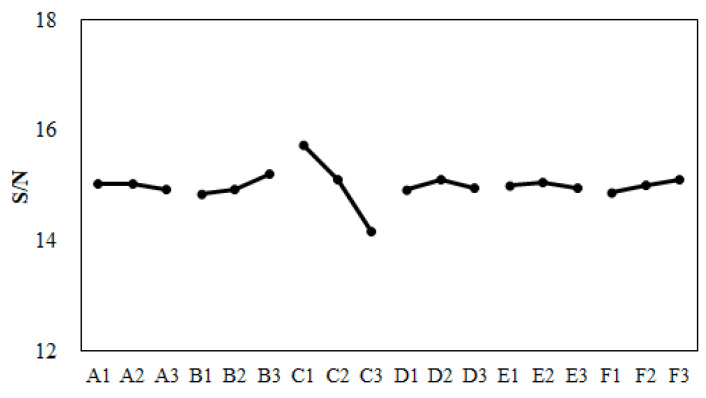
Response plot for various control factors after machine calibration.

**Figure 18 polymers-13-03065-f018:**
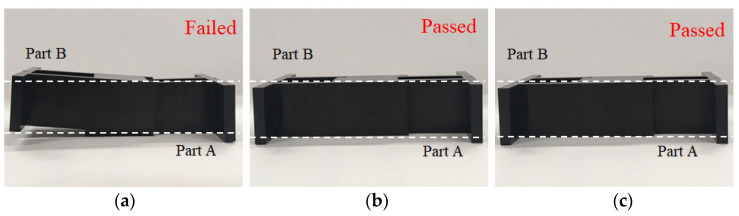
Experimental validation for the degree of assembly through the real integration test for CAE-DOE: (**a**) original design, (**b**) optimization before machine calibration, (**c**) optimization before machine calibration.

**Table 1 polymers-13-03065-t001:** The operation conditions for the single factor test.

Factor	Level 1	Level 2	Level 3	Level 4	Level 5
Injection speed ^a^ (%)	30	40	50	60	70
Melt temperature (°C)	190	200	210	220	230
Mold temperature (°C)	30	40	50	60	70
Packing time (s)	3	5	7	9	11
Packing pressure ^b^ (%)	50	75	100	125	150
Cooling time (s)	7	9	11	13	15

^a^: based on the maximum speed of the screw movement with 125 mm/s. ^b^: based on the end of filling pressure, P_EOF_.

**Table 2 polymers-13-03065-t002:** The key operation parameters utilized in the literature.

Ref.	Authors	Year	Key Operation Parameters							
			Mold Temp.	Melt Temp.	Injection Speed/Time	Injection Pressure	Packing Pressure	Packing Time	Cooling Temp.	Cooling Time
14	Lee and Kim	1995		✓	✓		✓	✓	✓	
15	Leo and Cuvelliez	1996					✓			
16	Yen et al.	2006								
17	Zhai et al.	2009	✓	✓		✓				
18	Othman et al.	2013	✓	✓	✓		✓			✓
21	Kova and Solymossy	2009	✓	✓						
22	Hakimian and Sulong	2012	✓	✓				✓		✓
23	Ozcelik and Erzurumlu	2006	✓	✓			✓	✓		✓
24	Zhai and Xie	2010								
25	Chiang and Chang	2007	✓				✓	✓		✓
27	Tsai and Tang	2014	✓	✓	✓		✓	✓		✓
28	Xu and Yang	2015	✓	✓	✓		✓	✓		✓
29	Kitayama et al.	2018		✓	✓		✓	✓	✓	✓
30	Hentati	2019	✓	✓		✓		✓		
31	Huang et al.	2020	✓	✓	✓		✓	✓		✓

Where “✓” means that parameter has been considered in that reference.

**Table 3 polymers-13-03065-t003:** Process conditions for the basic test.

Factor	Operation Conditions
Injection speed (%)	50
Melt temperature (°C)	210
Mold temperature (°C)	50
Packing time (s)	7
Packing pressure (%)	25; 50; 75; 100
Cooling time (s)	11

**Table 4 polymers-13-03065-t004:** The control factors and their levels in CAE-DOE before machine calibration.

Control Factor		Level 1	Level 2	Level 3
A	Injection Speed (mm/s)	25 (20%)	75 (60%)	125 (100%)
B	Mold Temperature (°C)	30	50	70
C	Packing Pressure (MPa)	95	126	158
D	Packing Time (s)	5	7	9
E	Melt Temperature (°C)	200	210	220
F	Cooling Time (s)	9	11	13

**Table 5 polymers-13-03065-t005:** L_18_(2^1^ × 3^7^) orthogonal array for CAE-DOE performance.

Exp		A	B	C	D	E	F	
		Injection Speed (mm/s)	Mold Temp. (°C)	Packing Pressure (MPa)	Packing Time (s)	Melt Temp. (°C)	Cooling Time (s)	
	**1**	**2**	**3**	**4**	**5**	**6**	**7**	**8**
**1**	1	1	1	1	1	1	1	1
**2**	1	1	2	2	2	2	2	2
**3**	1	1	3	3	3	3	3	3
**4**	1	2	1	1	2	2	3	3
**5**	1	2	2	2	3	3	1	1
**6**	1	2	3	3	1	1	2	2
**7**	1	3	1	2	1	3	2	3
**8**	1	3	2	3	2	1	3	1
**9**	1	3	3	1	3	2	1	2
**10**	2	1	1	3	3	2	2	1
**11**	2	1	2	1	1	3	3	2
**12**	2	1	3	2	2	1	1	3
**13**	2	2	1	2	3	1	3	2
**14**	2	2	2	3	1	2	1	3
**15**	2	2	3	1	2	3	2	1
**16**	2	3	1	3	2	3	1	2
**17**	2	3	2	1	3	1	2	3
**18**	2	3	3	2	1	2	3	1

**Table 6 polymers-13-03065-t006:** Difference of the characteristic lengths between the 100% and 25% packing pressure settings for both simulation and experiment (unit: mm).

	ΔX_1_	ΔX_2_	ΔX_3_	ΔX_4_	Ave ΔX
Simulation	0.032	−0.031	−0.065	−0.084	−0.037
Experiment	0.077	−0.027	−0.160	−0.135	−0.061

**Table 7 polymers-13-03065-t007:** Matched pairs of injection speed settings for the simulation and experiment systems.

Simulation	Simulation Injection Speed Setting (mm/s)	Experiment Injection Speed Setting ^a^
90%	67.0	30%
100%	73.4	40%
110%	79.5	50%
120%	85.8	60%
130%	92.0	70%

^a^: based on the maximum speed of the screw movement with 125 mm/s.

**Table 8 polymers-13-03065-t008:** Measurements of the calibration effect for the 50% injection speed setting at the 100% packing pressure setting.

	before Calibration			after Calibration			Calibration Rate (%)
Character. Length	Sim	Exp	ΔL	(Sim)cal	(Exp)cal	(ΔL)cal	
X_1_	−0.175	0.042	0.217	−0.170	0.042	0.212	2
X_2_	−0.255	−0.267	−0.012	−0.260	−0.267	−0.007	42
X_3_	−0.080	−0.167	−0.087	−0.096	−0.167	−0.071	18
X_4_	−0.103	−0.280	−0.177	−0.119	−0.280	−0.161	9
Average calibration rate							18

**Table 9 polymers-13-03065-t009:** Measurement of the calibration effect for the 50% injection speed setting at various packing pressure settings.

	Machine Calibration Rate (%)				
Packing Pressure Setting (%)	X_1_	X_2_	X_3_	X_4_	Average
25	0	0	9	0	2
50	1	0	11	8	5
75	2	42	14	8	17
100	2	42	18	9	18
Total average calibration rate					10

**Table 10 polymers-13-03065-t010:** Quantification of the degree of assembly through the integration test for the 50% injection speed system, experimentally.

Packing Pressure (%)	X_1_	X_2_	X_3_	X_4_	Integration Test
25	−0.035	−0.240	−0.007	−0.145	passed
50	−0.008	−0.233	−0.120	−0.173	passed
75	0.017	**−0.250**	−0.152	−0.238	passed
100	0.042	−0.267	−0.167	−0.280	failed

**Table 11 polymers-13-03065-t011:** Quantification the degree of assembly for the 50% injection speed setting in the simulation system.

Packing Pressure (%)	X_1_	X_2_	X_3_	X_4_	Integration Test
25	−0.207	−0.224	0.003	−0.013	passed
50	−0196	−0.235	−0.028	−0.045	passed
75	−0.187	**−0.243**	−0.050	−0.069	passed
100	−0.170	−0.260	−0.096	−0.119	failed

**Table 12 polymers-13-03065-t012:** Quality predictions based on the characteristic length for CAE-DOE before machine calibration.

Exp	Characteristic Lengths (mm)				Average	Sn	S/N
	X1	X2	X3	X4			
1	−0.15	−0.25	−0.08	−0.12	−0.15	0.06	15.82
2	−0.16	−0.27	−0.11	−0.14	−0.17	0.06	14.93
3	−0.13	−0.29	−0.20	−0.22	−0.21	0.06	13.27
4	−0.20	−0.24	−0.04	−0.06	−0.14	0.09	15.88
5	−0.15	−0.27	−0.13	−0.15	−0.17	0.05	14.80
6	−0.15	−0.28	−0.15	−0.19	−0.19	0.05	13.95
7	−0.15	−0.27	−0.14	−0.17	−0.18	0.05	14.52
8	−0.14	−0.28	−0.19	−0.20	−0.20	0.05	13.53
9	−0.17	−0.25	−0.09	−0.11	−0.15	0.06	15.54
10	−0.12	−0.29	−0.21	−0.23	−0.21	0.06	13.22
11	−0.18	−0.25	−0.07	−0.09	−0.15	0.07	15.74
12	−0.15	−0.26	−0.11	−0.15	−0.17	0.06	14.99
13	−0.16	−0.26	−0.14	−0.14	−0.17	0.05	14.88
14	−0.11	−0.29	−0.21	−0.25	−0.21	0.07	12.99
15	−0.18	−0.24	−0.06	−0.08	−0.14	0.08	15.93
16	−0.12	−0.29	−0.20	−0.24	−0.21	0.06	13.09
17	−0.15	−0.26	−0.16	−0.16	−0.18	0.05	14.50
18	−0.18	−0.26	−0.10	−0.13	−0.17	0.06	15.00

**Table 13 polymers-13-03065-t013:** Response values for various control factors before machine calibration.

	A	B	C	D	E	F
Level 1	14.66	14.57	15.57	14.67	14.61	14.54
Level 2	14.74	14.42	14.85	14.72	14.60	14.51
Level 3	14.36	14.78	13.34	14.37	14.56	14.72
E_i_^1−2^	0.08	−0.15	−0.72	0.05	−0.02	−0.03
E_i_^2−3^	−0.37	0.36	−1.51	−0.36	−0.04	0.21
Range	0.37	0.36	2.23	0.36	0.05	0.21
Rank	2	3	1	4	6	5

Where E_i_^1−2^ means the influence of the “i” factor on the S/N ratio from Level 1 to Level 2; E_i_^2−3^ means the influence of the “i” factor on the S/N ratio from Level 2 to Level 3.

**Table 14 polymers-13-03065-t014:** Control factors and their levels in CAE-DOE after machine calibration.

Control Factor		Level 1	Level 2	Level 3
A	Injection Speed (mm/s)	67	79.5	92
B	Mold Temperature (°C)	30	50	70
C	Packing Pressure (MPa)	90	120	140
D	Packing Time (s)	5	7	9
E	Melt Temperature (°C)	200	210	220
F	Cooling Time (s)	9	11	13

**Table 15 polymers-13-03065-t015:** Quality predictions based on the characteristic lengths for CAE-DOE after machine calibration.

Exp	Characteristic Lengths (mm)				Average	Sn	S/N
	X1	X2	X3	X4			
1	−0.14	−0.26	−0.10	−0.14	−0.16	0.06	15.41
2	−0.17	−0.26	−0.10	−0.12	−0.16	0.06	15.20
3	−0.15	−0.27	−0.16	−0.17	−0.19	0.05	14.19
4	−0.20	−0.24	−0.04	−0.05	−0.13	0.09	15.94
5	−0.15	−0.26	−0.11	−0.14	−0.17	0.06	15.08
6	−0.16	−0.27	−0.12	−0.16	−0.18	0.06	14.62
7	−0.15	−0.27	−0.13	−0.16	−0.18	0.05	14.73
8	−0.16	−0.27	−0.14	−0.16	−0.18	0.05	14.46
9	−0.18	−0.24	−0.06	−0.07	−0.14	0.08	16.01
10	−0.14	−0.27	−0.16	−0.18	−0.19	0.05	14.21
11	−0.19	−0.24	−0.05	−0.08	−0.14	0.08	15.88
12	−0.17	−0.25	−0.10	−0.13	−0.16	0.06	15.33
13	−0.16	−0.26	−0.13	−0.13	−0.17	0.05	15.01
14	−0.12	−0.28	−0.18	−0.22	−0.20	0.06	13.68
15	−0.18	−0.24	−0.10	−0.02	−0.14	0.08	15.94
16	−0.12	−0.28	−0.17	−0.21	−0.19	0.06	13.85
17	−0.17	−0.25	−0.11	−0.11	−0.16	0.06	15.30
18	−0.18	−0.26	−0.09	−0.11	−0.16	0.07	15.25

**Table 16 polymers-13-03065-t016:** Response values for various control factors after machine calibration.

	A	B	C	D	E	F
Level 1	15.04	14.86	15.75	14.93	15.02	14.89
Level 2	15.04	14.93	15.10	15.12	15.05	15.00
Level 3	14.94	15.22	14.17	14.97	14.95	15.12
E_i_^1−2^	0.00	0.08	−0.65	0.19	0.03	0.10
E_i_^2−3^	−0.11	0.29	−0.93	−0.15	−0.10	0.12
Range	0.11	0.36	1.58	0.19	0.10	0.23
Rank	5	2	1	4	6	3

## Data Availability

The data presented in this study are available on request from the corresponding author.

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
