# Peer review of "A Methodology to Predict and Optimize Ease of Assembly for Injected Parts in a Family-Mold System"

_polymers, 2021, doi:10.3390/polym13183065_

Round 1

Reviewer 1 Report

I would like to thank the Authors for dedicating time and effort to reviewing the paper. Unfortunately, I do not think that the manuscript holds enough scientific value and quality to recommend its publication. The submission lacks scientific hypotheses, the experimental and simulation approaches are confusing, and the quality of presentation does not match the high-quality standard of scientific journals. The topic can be of interest, however, I recommend rethinking the whole approach to experimentation by starting with more clearly defined goals.

Author Response

Dear Reviewer,

Thank you very much for your patience and suggestion.  Regarding your comments, we have tried the best to answer as described in the attached file. 

Thank you very much.

Chao-Tsai Huang 

Reviewer 2 Report

In the manuscript “A Methodology to Predict and Optimize the Ease of the Assembly for the Injected Parts in a Family-mold System”, the authors studied the assembly behavior of  two injected components made by a family mold system  using numerical simulation and experimental validation. Please, consider the following recommendations to improve your document:

  1. Modify the abstract, concentrating the most relevant information. For example, lines 14 to 20 are a justification of the work that can move to the introduction.
  2. Information about the objective of the work is presented in the abstract and at the end of the introduction. Usually, the goal only goes to the end of the introduction.
  3. About material, ABS 144 (PA757 supplied by Che-Mei). Please add city and country where it was purchased.
  4. Indicate how the viscosity data and PVT curves for ABS (Figure 3) were measured or obtained
  5. Remove the box from the figures 6, 8, 9, 11, 13, 14, 16, 17

Author Response

Dear Reviewer,

Thank you very much for your patience to give us the priceless suggestions and comments.  Regarding those suggestions and comments we have tired the best to answer one-by-one.  Please see the attachment.

Thank you very much and Best Regards,

Chao-Tsai Huang

Reviewer 3 Report

This paper presents an analysis of the flow and shrinkage behavior of an injection molding system based on a combination of DOE and CAE-DOE. The paper is well written, however , I have the following concerns : 
1. The main contribution of this paper is to compare the performance of the simulation and the machine calibration of the system . It would be better if the authors could provide a more detailed analysis of what is the difference between the simulation performance and the calibration performance. 
2. It would be more convincing if the author can provide a comparison between the results of the single test and the integration test. 
3. It is not clear to me how the contribution of the proposed method is different from that of Huang et al . [ 33 ] . 
4 .The authors should provide more details about the setup of the machine . For example , how many parameters are used in the simulation ? What is the number of parameters in the machine ? How many parameters were used for the calibration ? 

Author Response

Dear Reviewer,

Thank you very much for your patience to provide us the wonderful suggestions and comments.  Regarding those suggestions and comments we have tired the best to answer one-by-one.  Please see the attachment.

Thank you very much and Best Regards,

Chao-Tsai Huang

Round 2

Reviewer 1 Report

I previously indicated in my report that the paper should be rejected.